# Exploring Pt-Impregnated CdS/TiO_2_ Heterostructures for CO_2_ Photoreduction

**DOI:** 10.3390/nano14221809

**Published:** 2024-11-12

**Authors:** Lidia García-Santos, Javier Fernández-Catalá, Ángel Berenguer-Murcia, Diego Cazorla-Amorós

**Affiliations:** Inorganic Chemistry Department, Materials Science Institute, University of Alicante, Ap. 99, 03080 Alicante, Spain; lidia.garciasantos@ua.es (L.G.-S.); j.fernandezcatala@ua.es (J.F.-C.); cazorla@ua.es (D.C.-A.)

**Keywords:** carbon dioxide, photocatalysis, cadmium sulfide, titania P25, Pt, methane

## Abstract

This work focuses on the production of methane through the photocatalytic reduction of carbon dioxide using Pt-doped CdS/TiO_2_ heterostructures. The photocatalysts were prepared using P25 commercial titania and CdS synthesized through a solvothermal methodology, followed by the impregnation of Pt onto the surface to enhance the physicochemical properties of the resulting photocatalysts. The pure and heterostructure-based materials were characterized using X-ray diffraction (XRD), inductively coupled plasma optical emission spectroscopy (ICP-OES), scanning electron microscopy (SEM) with energy dispersive X-ray spectroscopy (EDX), transmission electron microscopy (TEM), X-ray photoelectron spectroscopy (XPS), ultraviolet-visible spectroscopy (UV-Vis), ultraviolet photoelectron spectroscopy (UPS), and photoluminescence spectroscopy (PL). The obtained results show the successful synthesis of the heterostructure impregnated with Pt. Moreover, the observed key role of CdS and Pt nanoparticles in the final semiconductor is to reduce the electron-hole pair recombination rate by acting as an electron sink, which slows down the recombination process and increases the photocatalyst efficiency. Thus, Pt-doped CdS/TiO_2_ heterostructures with the best observed composition presents better catalytic activity than P25 titania with methane production values being 460 and 397 µmol CH_4_/g·h, respectively.

## 1. Introduction

The Industrial Era, recognized as one of the most significant technological, economic, and social revolutions in modern history, spanned from the mid-18th century to the 20th century [1]. Since then, the current energy landscape has been profoundly shaped by the availability, stability, and high energy density of fossil fuels such as oil, natural gas, and coal. Consequently, these fuels have become the primary source of conventional energy used worldwide [2]. However, meeting the global energy demand of an ever-growing society through an energy model based on burning fossil fuels has a negative impact on the environment [3]. Anthropogenic carbon dioxide is the main greenhouse gas emitted directly into the atmosphere during the combustion of fossil fuels [4]. Despite increasing global prosperity, global CO_2_ emissions are experiencing a structural slowdown due to the rising use of renewable energies [5]. Nonetheless, this progress is not enough to fully counteract one of the most significant consequences of rising CO_2_ emissions and their accumulation in the atmosphere: global warming [6].

Thanks to its physicochemical properties, carbon dioxide conversion can contribute to the development of products and services with a reduced carbon footprint. Thus, this gas becomes a valuable resource for a wide range of applications [7,8,9]. Among the various alternatives for carbon dioxide utilization, the production of fuels and high-value-added products emerges as a particularly relevant approach in the current scenario. To this end, several pathways can be considered, including thermal [10], electrochemical [11], or biological reduction (via natural photosynthesis) [12] and photochemical, also known as artificial photosynthesis or CO_2_ photoreduction [13]. Among the multiple proposed options, artificial photosynthesis stands out due to its potential to harness solar energy which is abundant, economical, and environmentally clean and safe [14]. Artificial photosynthesis involves a process of heterogeneous photocatalysis. In this way, this technology is capable of generating high-value-added products and reducing CO_2_ emissions using solar radiation, water, and a semiconductor [13,15,16,17].

Since 1972, when Fujishima and Honda reported that titanium dioxide (TiO_2_) could induce the photocatalytic dissociation of water when exposed to sunlight, this material has become the most widely used photocatalyst in environmental applications, such as artificial photosynthesis [18]. This is due to several desirable properties, including its natural abundance, high resistance to photo-induced corrosion, chemical stability, and environmental sustainability [19]. Commercially, the most commonly studied form of titanium dioxide is P25 (Degussa, Frankfurt, Germany), which is composed of a mixture of Anatase (around 70%) and Rutile (around 30%) phases, but it seems that the exact crystalline composition is not known [20]. Its key properties include a crystal size between 20–50 nm, a surface area of approximately 50 m^2^/g, and a band gap of 3.1 eV. These characteristics make the commercial P25 material highly suitable for photocatalytic applications, such as the photoreduction of carbon dioxide [19,21]. However, P25 has certain drawbacks, including low surface area, limited visible light absorption, and a high rate of electron-hole recombination that negatively affects its photocatalytic activity [22]. Several properties of the catalyst can be modified or improved to enhance its catalytic performance in this process.

In recent years, the scientific community has made significant efforts to improve the properties of photocatalysts for the photoreduction of carbon dioxide. Several approaches have been explored [23], including heteroatom doping [24], structural design, crystallinity modification, surface defect engineering [25,26], and the formation of heterostructures with other semiconductors or co-catalysts [27]. Among these, the formation of heterostructures is considered one of the most viable options to improve the catalytic efficiency of P25 titania. Given that P25 has a high oxidation potential, it is typically paired with a semiconductor that has the highest possible reduction potential (in absolute terms), such as CdS, which is an n-type intrinsic semiconductor with a band gap of around 2.4 eV [28,29]. Additionally, adding co-catalysts such as Pt nanoparticles, which is one of the most promising noble metals used in this area, reduces the electron-hole pair recombination rate, which slows down the recombination process and increases the photocatalyst efficiency [30,31]. Wei et al. have reported the synthesis of 3D ordered macroporous TiO_2_-supported Pt@CdS core–shell nanoparticles by the gas bubbling-assisted membrane reduction-precipitation (GBMR/P) method [32]. Other reported works based on elaborated studies of the electronic interactions on Pt and CdS-(Metal-Oxide)-based photocatalysts obtained promising results [33,34,35]. However, the scientific community agrees that it is necessary to develop photocatalysts with higher catalytic activity for their commercialization [36,37,38].

In this work, the effect of modifying P25 titania with other semiconductors like CdS, in order to create a heterostructure, is investigated. This was followed by the addition of Pt species as co-catalysts. Thus, the strategy for constructing interfaces between two semiconductors plays a crucial role in the photocatalytic performance of the resulting systems, as the quality of the interface is determined by the synthesis method [39]. For this reason, facile solvothermal synthesis was used to prepare CdS-based photocatalysts supported on TiO_2_ [40], with Pt nanoparticles incorporated through an impregnation method [41,42], to produce methane from the photoreduction of carbon dioxide in the presence of water vapor.

## 2. Materials and Methods

### 2.1. Materials

The reactants used in this work (without any further purification) were commercial TiO_2_ (P25, Rutile–Anatase 85:15, 99.9%, Degussa), cadmium nitrate tetrahydrate (Cd(NO_3_)_2_·4H_2_O, 98%, Sigma Aldrich, St. Louis, MO, USA), anhydrous sodium thiosulfate (Na_2_S_2_O_3_, 99%, Sigma Aldrich), polyvinylpyrrolidone PVP10-100G ((C_6_H_9_NO)_n_, Sigma Aldrich), ethylenediamine (C_2_H_8_N_2_, 99%, Sigma Aldrich), hexachloroplatinic hexahydrate acid (H_2_PtCl_6_·6H_2_O, 37.5%, Sigma Aldrich), sodium borohydride (NaBH_4_, 98%, Sigma Aldrich), hydrochloric acid (HCl, 37%, Fisher Scientific, Waltham, MA, USA), nitric acid (HNO_3_, 60%, VWR Chemicals), absolute ethanol (EtOH, ≥99.8%, VWR CHEMICALS, Radnor, PA, USA), and deionized water. Moreover, glass plates (24 × 24 mm, Thermo Scientific Menzel, Fisher Scientific) were used.

### 2.2. Synthesis of CdS/TiO_2_ Photocatalysts

For the synthesis of cadmium sulfide, Cd(NO_3_)_2_·4H_2_O and anhydrous Na_2_S_2_O_3_ were used as Cd and S salt precursors, respectively. CdS was prepared in the presence of P25 titania through a solvothermal method using ethylenediamine as the reaction medium [40]. A scheme of the synthesis method is presented in Appendix A.

As an illustrative example, the synthesis of CdS material on titania P25 was performed as follows: Cd(NO_3_)_2_·4H_2_O and anhydrous Na_2_S_2_O_3_ salts along with the surfactant PVP were dissolved in 40 mL of ethylenediamine. PVP prevents agglomeration of the synthesized particles and protects them, acting as a capping agent [41]. The amount of CdS in the synthesized catalysts was modified as shown in Table 1. Then, TiO_2_ P25 was added, and the mixture was vigorously stirred for 15 min using a magnetic stirrer. Afterward, the dispersion was transferred into an ultrasonic bath to achieve a homogenous solution with uniformly dispersed P25 particles. The mixture was maintained at room temperature in the ultrasonic bath for 15 min. Upon completion, the solution was poured into a 50 mL Teflon container inside a steel autoclave which was placed in an oven at 180 °C for 24 h. After this time, the solution was filtered using a vacuum filtration system and thoroughly washed three times with a 1:1 mixture of water and ethanol. Subsequently, the sample was dried in an oven at 60 °C for 24 h. Finally, the material was ground with an agate mortar to obtain a fine powder. The samples prepared were named Cd(X)-Ti, depending on the content of CdS (X).

### 2.3. Synthesis of Pt/CdS/TiO_2_ Photocatalysts

The impregnation of Pt on the surface of CdS/TiO_2_ materials was carried out by a modification of a standard impregnation protocol followed by the reduction of a platinum precursor such as H_2_PtCl_6_·6H_2_O with NaBH_4_ [42,43]. A scheme of the synthesis method is presented in Appendix A.

As an illustrative example, the synthesis of Pt NPs on heterostructures was performed as follows: First of all, 0.5 g of the Cd(X)-Ti sample were dispersed in 50 mL of deionized water using an ultrasonic bath for 15 min. At the same time, a 6.6 mM solution of H_2_PtCl_6_·6H_2_O in water was prepared. To achieve a photocatalyst containing 1%wt. in Pt, the volume of solution pipetted was 16.2 mL. After 15 min, this solution was added to the suspension of the CdS/TiO_2_ material in water, and the mixture was kept under agitation for one day. The following day, the reducing agent solution, specifically NaBH_4_, was prepared. To achieve this, 0.1 g of the reducing agent was dissolved in 20 mL of water. The molar ratio of Pt to NaBH_4_ is 1:0.5. Consequently, the volume of the NaBH_4_ solution to be pipetted was 0.98 mL. The NaBH_4_ solution was added dropwise to the previous solution under continuous stirring, resulting in a color change from yellow to light gray. The mixture was then left to stir for an additional 2 h. After this time, the solution was filtered using a vacuum filtration system and thoroughly washed three times with a 1:1 mixture of water and ethanol. Subsequently, the sample was returned to the oven at 60 °C for 24 h in order to dry the solid. Finally, the material was grounded with a mortar to obtain a fine powder.

The CdS/TiO_2_ photocatalysts were labeled as Cd(X)-Ti, and those containing Pt were labeled as Pt/Cd(X)-Ti, where “Cd” stands for CdS, “X” represents its percentage in the photocatalyst, and “Ti” refers to TiO_2_. Table 2 provides labelled samples that will be studied, as well as for CdS and TiO_2_-P25, separately.

### 2.4. Characterization Techniques

The crystallinity of the samples was determined by X-ray diffraction (XRD) using a Rigaku Miniflex II X-ray diffractometer. Diffractograms were recorded using CuKα radiation (λ = 0.15418 nm) over a 2θ range from 5° to 60°, with a step size of 0.05°, and analyzed using HighScore v4.9. software [44]. The crystallite size was estimated by applying the Scherrer equation [45]. To subtract the instrumental broadening factor from the measured FWHM value, a crystalline SiC pattern was used. The corrected FWHM value of the sample was calculated according to the following Equation (1):(1)$$\beta_{real}^{2}=\beta_{obs}^{2}-\beta_{inst}^{2}$$
where *β_real_* is the value obtained from the contribution of the crystallite size, β*_obs_* is the measured value, and β*_inst_* is the broadening related to the instrument [45].

The platinum content in Pt/Cd(X)-Ti materials was measured in an Optima 7300 DV inductively coupled plasma optical emission spectrometer (ICP-OES) with dual view from Perkin Elmer. The preparation of solutions of the different catalysts was carried out by treating them with aqua regia for 48 h under stirring at room temperature in order to lixiviate all the metal present in the catalyst.

Compositional microanalysis and elemental mapping of the Pt/Cd(20)-Ti sample were conducted using a JEOL IT500HR/LA (JEOL, Tokyo, Japan) high-resolution scanning electron microscope (SEM). This microscope is equipped with an EDX analysis system.

The morphology analysis and distribution of Pt nanoparticles were analyzed by a JEOL JEM-2010 transmission electron microscope (TEM) with a GATAN ORIUS SC600 camera and GATAN Digital Micrograph 1.80.70 acquisition (GATAN, Pleasanton, CA, USA) and image processing software for GMS 1.8.0. The particle size study was conducted by measuring 100 particles using ImageJ v1.54k software [46].

The light absorption of the different materials was analyzed using a JASCO V-670 UV-Vis spectrophotometer (JASCO Spain, Madrid, Spain), equipped with a dual grating monochromator, a detector consisting of photomultiplier tubes, and light sources including a deuterium lamp (190 to 350 nm) and a halogen lamp (330 to 2700 nm). BaSO_4_ was used as the reference material. The absorption edge wavelength was estimated from the intercept at zero absorbance of the high slope portion of each spectrum in the range 200–800 nm (absorbance method) [47]. Then, the band gap can be calculated, following the Equation (2):(2)$$E_{g}=\frac{1239.8}{\lambda }$$
where *E_g_* is the bandgap energy (eV), 1239.8 (eV·nm) is the resulting product of Planck constant (eV·s) and the speed of light (nm·s^−1^), and *λ* (nm) is the wavelength of the absorption edge [48]. While the Tauc plot may be considered a more accurate procedure to obtain band gap values, the nature of the optical transitions (direct or indirect transitions) in the semiconductor would need to be elucidated by calculating Tauc plot functions. In this respect, Tauc plot functions vary depending on the transitions. For comparative purposes, the band gaps of the materials were also calculated by the Tauc plot method. In the case of CdS (a direct band gap material), the obtained results in both cases were similar to the values obtained using the absorbance method. However, the P25 semiconductor, being composed of a mixture of Anatase (direct band gap) and Rutile (indirect band gap), rendered the Tauc plot method as inaccurate for this composite. For this reason, we opted to use the absorbance method in order to estimate the band gap of all materials for the sake of accuracy and comparability [46].

Steady-state photoluminescence (PL) spectra of samples were recorded on a PicoQuant Fluotime 250 device, Berlin Germany (Fluorescence LifeTime Spectrometer) using an excitation wavelength of 375 nm by means of a diode Laser (LDH-P-C-375) at room temperature. The time-resolved PL spectra were obtained on the same fluorescence lifetime spectrometer with a diode laser drive. Two peaks were analyzed using decay curves in a maximum peak between 350–450 nm and 500–600 from TiO_2_ and CdS, respectively. They were fitted using the exponential Tail Fit model, as described in Equation (3).
(3)$$Dec\left(t\right)=\sum_{i=1}^{n_{Exp}}A_{i}\;e^{-\frac{\tau }{\tau_{i}}}+{Bkgr}_{Dec}$$

By using this Equation and solving for t, the $A_{i}$ and $\tau_{i}$ are obtained, which are then substituted into Equation (4) to calculate the average lifetime.
(4)$$t_{Av\;Amp}=\sum_{\begin{array}{c} i=1\\ Ai>0\; \end{array}}^{n_{Exp}}A_{i}\tau_{i}\;\;\;/\;\sum_{\begin{array}{c} i=1\\ Ai>0\; \end{array}}^{n_{Exp}}A_{i}\;\;\;$$

The average lifetime of the charge carrier in the photocatalysts was determined using Equation (4) and fitting the time-resolved PL spectra into three lifetimes.

An XPS of Pt/Cd(X)-Ti and pure (TiO_2_ and CdS) materials were tested using an automated K-Alpha spectrometer from Thermo Scientific with a high-resolution monochromator. The XPS instrument was equipped with a monochromatic Al Kα X-ray source (1486.6 eV) operating at 100 W. The alpha hemispherical analyzer was operated in the constant energy mode with survey scan pass energies of 200 eV to measure the whole energy band and 50 eV in a narrow scan to selectively measure the specific elements. All peaks were calibrated using the C1s peak binding energy of 284.6 eV. XPS data were analyzed with Avantage v5.9929 software [49] and the experimental curves were fitted using Lorentz–Gaussian functions, and the background was a Shirley curve.

Ultraviolet photoelectron spectroscopy (UPS) was used in order to study the position of the valence band (VB) of the synthesized semiconductors. UPS spectra were recorded on a fully automated Thermo-Scientific NEXSA multi-surface analysis system, capable of acquiring spectra with a high-resolution monochromator. It is equipped with an automated charge compensation system with electron and ion sources, as well as an argon ion source for high-precision sputtering.

### 2.5. Catalytic Tests

Catalytic tests for methane production from the reduction of carbon dioxide in the presence of water vapor were carried out in a flow reactor under ambient conditions, as shown in Figure 1, that uses a thin layer of the catalyst. The sample analysis was performed using gas chromatography (GC) in an Agilent 6890 N chromatograph, equipped with a CTR-I column to separate products, by-products, and reactants, using two detectors: a flame ionization detector (FID) and a thermal conductivity detector (TCD). The catalytic activity of all materials listed in Table 2 was evaluated. Each gas stream coming out from the reactor was sampled every 10 min in the GC during 4 h of reaction.

The procedure is as follows: 40 mg of the catalyst are spread on the surface of a glass plate (24 × 24 mm optical microscopy coverslip) to form a thin layer. The glass plate is placed inside a stainless-steel reactor, that includes a quartz window, designed to operate in a continuous flow regime. The system is purged using pure He (99.999%) with a flow rate of 5 mL/min passing through the reactor. Additionally, water vapor is introduced into the pure He stream using a bubbler containing 2.5 mL of water and 100 µL of ethanol (30 ± 2.0 °C corresponding to a water vapor pressure of 0.0418 bar) to saturate the He stream with H_2_O until a stable state is reached (12 h).

After this time, the He flow is changed to a flow of 5 mL/min of pure CO_2_ (99.999%) under the same previously described conditions (i.e., passed through the bubbler with water) until a steady state is reached again (1 h). Once the steady state is achieved and before starting the catalytic test, the chromatograph sequence is initiated (corresponding to an analysis every 10 min) in the absence of light for 30 min.

Finally, the catalyst was irradiated with a UV LED light with a wavelength of 368 nm at room temperature. The working conditions for CO_2_ reduction to CH_4_ were a working current intensity of 1.5 A and a voltage of 3.75 V (5.6 W) at room temperature. The presence of other products, such as higher hydrocarbons, specifically acetaldehyde resulting from ethanol oxidation, was also analyzed during photoreaction. There were other subproduct traces that could not be quantified with this set up.

## 3. Results and Discussion

### 3.1. Characterization Results

The XRD patterns of all photocatalysts are shown in Figure 2 and Table 3 and contain information about the crystalline phases detected. The results (Figure 2) showed that pure P25 (commercial TiO_2_ obtained from Degussa) is composed by both Rutile (JCPDS reference number 96-900-4142) and Anatase (JCPDS reference number 96-152-6932) phases (marked as a blue and yellow rhombus in the XRD diffractogram for Rutile and Anatase, respectively), with a ratio of 85:15, according to the characteristics specified by the supplier; and pure CdS comprises both Wurtzite (hexagonal; JCPDS reference number 96-101-1055) and Sphalerite (cubic; JCPDS reference number 96-101-1252) phases (marked as a green and purple rhombus in the XRD diffractogram for Wurtzite and Spharelite, respectively). Cd(X)-Ti and Pt/Cd(X)-Ti materials (marked as black and red patterns, respectively) have the typical characteristic diffraction patterns of all four crystalline phases mentioned above (Figure 2 and Table 4). Furthermore, it is important to note that the XRD pattern of the samples impregnated with Pt (Pt/Cd(X)-Ti samples) did not show any signals related to the presence of Pt species (XRDs are plotted in red lines), likely due to the low amount of platinum and the good dispersion of the nanoparticles on the surface of the materials.

The XRD patterns of Cd(X)-Ti and Pt/Cd(X)-Ti photocatalysts are similar, but the intensity of some of the CdS peaks are lower after Pt impregnation. This suggests that the content of CdS and its crystalline phases decrease after the impregnation process. For this reason, the percentage of crystalline phase present in the materials was determined. The percentage of crystalline phase, calculated by the Rietveld refinement method provided by the HighScore software (see Table 4), confirms this hypothesis. This effect could be attributed to the reagents used during the impregnation process, which may cause a partial degradation of the material.

Table 5 displays the crystallite average particle size of CdS and TiO_2_ which were calculated from the full width at half maximum (FWHM) of the main intensity peak of the dominant crystalline phases (Wurtzite and Anatase in case of CdS and TiO_2_, respectively). However, it was not possible to calculate the crystal size for the CdS phase in Cd(1)-Ti and Pt/Cd(1)-Ti materials due to the low content of this phase. The results show a decrease in the crystal size of CdS in materials containing Pt. This effect could be attributed to the removal of part of the material during the impregnation process.

In order to verify that Pt was successfully loaded on photocatalysts, the Pt content was analyzed by ICP-OES (Table 6). The results indicate that the amount of platinum varies depending on the CdS content, being higher with the CdS content. This may be attributed to a stronger interaction between Pt and CdS than with TiO_2_.

To visually confirm the successful combination of CdS, TiO_2_, and Pt, the Pt/Cd(20)-Ti material was analyzed by SEM, and the element distribution was performed by EDS. The results are shown in Figure 3 and Table 7. The EDS spectra is presented in Appendix A. Elemental mapping revealed that Ti and O elements are homogeneously distributed with an atomic ratio of 1:2, consistent with the stoichiometric formula of TiO_2_. In contrast, Cd and S elements have higher concentrations in specific areas, showing an atomic ratio of 1:1, consistent with the stoichiometric formula of CdS. Pt was also detected on the surface, though in very low amounts.

TEM images of CdS and Cd(20)-Ti materials before and after platinum impregnation were compared (Figure 4). Original CdS exhibits an irregular morphology, except for the presence of some short rod-shaped particles, as observed in Figure 4a,b. However, the morphology of this compound in the heterostructures is different, as seen mainly in Figure 4c,d, where the particles have a long rod shape. This phenomenon may be due to the presence of crystalline TiO_2_ that affects crystal growth. In the case of the heterostructures, cadmium sulfide crystals grow from already existing crystalline nuclei, namely TiO_2_ nanoparticles. Thus, CdS crystals with a more defined crystal habit and larger particle size are obtained. In the case of pure CdS, the absence of those TiO_2_ nuclei seems to promote the formation of many nuclei, resulting in smaller particle sizes. Regarding TiO_2_ P25, its morphology and particle size do not change in any case, matching the values found in the literature [20]. Therefore, it is deduced that the solvothermal synthesis does not alter this material. After Pt impregnation, the materials do not undergo important morphological changes. However, in Figure 4, it can be observed that after Pt incorporation, the particle contours are less defined (indicated in blue) and the particles exhibit more striations (indicated in red), suggesting that the material has been modified after impregnation. These results are in good agreement with XRD results.

Particle size distributions of the obtained platinum nanoparticles were further studied by TEM. Figure 5 shows micrographs of these nanoparticles (marked in red) on the different materials, and Table 8 lists their average sizes. The spheroidal particles with sizes ranging between 30 and 80 nm correspond to the commercial TiO_2_ nanoparticles. The results show a decrease in size of the metallic nanoparticles as the percentage of CdS increases in the photocatalysts. It appears that Pt interacts better with CdS, promoting the dispersion of Pt nanoparticles and reducing their aggregation during reduction. For the Pt/CdS material, it was not possible to determine the nanoparticle size due to their small size. Additionally, the thickness of the CdS particles does not facilitate the distinction of the supported nanoparticles.

Figure 6 presents the XPS spectra for Ti 2p, O 1s, Cd 3d, S 2p, and Pt 4f for the platinum containing materials.

The Ti 2p spectra (Figure 6a) shows two signals: Ti 2p_1/2_ (at higher binding energies) and Ti 2p_3/2_ (at lower binding energies). Peak deconvolution shows the presence of Ti^4+^ at a binding energy of 458.5 eV [53] in all samples except for the Pt/Cd(1)-Ti material. In this case, the signal is shifted to higher binding energy by +0.6 eV, suggesting a different chemical environment or atomic state. According to the literature, this signal could correspond to Ti_2_S_3_; however, sulfur was not detected on the surface of this sample, so it cannot be confirmed as a Ti-S interaction and may instead indicate a Ti^3+^-Ti^3+^ interaction [54]. This peak is also seen in the Pt/Cd(20)-Ti sample, where sulfur signals are more intense. Specifically, the S 2p_3/2_ peak appears at around 161.5 eV [55], which is typically associated with metal sulfides (Figure 6d). While distinguishing between Cd, Ti, and Pt, interaction with S is not possible, and an S-Metal interaction can be confirmed (Figure 6d).

The O 1s spectra (Figure 6b) shows signals corresponding to different types of bonds. These peaks reflect the variety of chemical environments of oxygen on the surface of Pt/Cd(X)-Ti materials, indicating interaction with titanium for signals at 529.5 and 531.7 eV, corresponding to possible Ti-O-Ti and Ti-O-H interactions, respectively [56]. In the Pt/Cd(1)-Ti photocatalyst, a shift in the most intense peak from the Ti-O-Ti signal is observed, which may be attributed to the previously mentioned interactions. Peaks assigned to C=O and C-O species (from lower to higher binding energy, respectively), are present due to the unavoidable presence of carbon from PVP surfactant, especially in the Pt/CdS sample [57].

The Cd 3d spectra (Figure 6c) reveals two signals corresponding to Cd 3d_3/2_ (at higher binding energies) and Cd 3d_5/2_ (at lower binding energies). Peak deconvolution indicates the presence of both coordinated and uncoordinated Cd^2+^ on the surface of the materials [58]. The formation of uncoordinated Cd^2+^ species may result from sulfur degradation during Pt impregnation, as CdS is a chalcogenide that undergoes photodegradation in water [59] and can be attacked by acids and reducing agents. This phenomenon is primarily observed in the Pt/Cd(1)-Ti sample, analyzed before and after Pt impregnation. Results show the presence of coordinated Cd^2+^ and sulfur (Appendix A), whereas after impregnation, predominantly uncoordinated Cd^2+^ and systematic absence of S are observed (Appendix A). This behavior decreases as the CdS concentration in the materials increases.

Finally, the Pt 4f spectra (Figure 6e) displays two signals corresponding to Pt 4f_7/2_ (at higher binding energies) and Pt 4f_5/2_ (at lower binding energies). Peak deconvolution indicates the presence of Pt species in various oxidation states: Pt^0^, Pt^2+^, and Pt^4+^ with binding energies of 71.3, 72.4, and 73.9 eV [60,61], respectively. The amount of CdS in the materials affects the formation of platinum species. According to the literature, oxidized Pt^2+^ and Pt^4+^ species have binding energies around 72.4 and 74.9 eV, respectively, while sulfur-bonded Pt^2+^ and Pt^4+^ species have binding energies around 72.3 and 73.9 eV [62]. The results suggest that Pt^4+^ is forming platinum (IV) sulfide species, although the amount of Pt^4+^ is very small in all of the cases, being predominant Pt^2+^ species for the catalysts with higher Cd contents (Figure 6f), which reflects the strong interaction with sulfur species that stabilizes this oxidation state.

The UV-Vis absorption properties of all materials were evaluated (Figure 7a). The Cd(X)-Ti samples prepared in this study exhibited strong UV absorption in the 250–400 nm range, clearly indicating the existence of Ti in octahedral coordination, which is characteristic of the P25 material. Additionally, they showed another absorption in the 450–600 nm range, characteristic of the CdS material. This behavior is also observed in the Pt/Cd(X)-Ti materials. The most significant changes are observed in the absorbance of the photocatalysts after impregnation, which decreases in all cases except for the Pt/P25 and Pt/Cd(1)-Ti samples. Notably, in these two photocatalysts, the color change after Pt impregnation was more intense compared to the other materials with a higher content of CdS (Figure 7b). In this case, materials that were initially white or pale yellow turned a grayish color indicative of the noble metal reduction. The remaining materials, which were yellow, darkened slightly. Consequently, light absorption occurs at different wavelengths for the materials with Pt. This phenomenon is evident in the results presented in Table 9, which also includes the band gap values for both pure and Pt-doped CdS and TiO_2_ P25 materials, calculated from the wavelength obtained by extrapolating the line with the steepest slope to zero absorbance.

To better establish the fine interplay between the components of the different photocatalysts, UPS spectra were collected for Pt/P25 and Pt/CdS samples and as well as for the parent P25 and CdS materials for comparison purposes. Prior to the analyses, the bias voltage was calibrated using a freshly cleaned Ag foil mounted on the sample holder. Figure 8 shows the obtained full UPS spectra of samples mentioned above. In the high-energy region of the spectra in Figure 8a (above 15 eV), a pronounced peak is observed for both cases, especially for the P25 spectrum. According to the literature [42], this sharp spike at high binding energies (or low kinetic energies) is due to the effect of the flood gun, which was used throughout all the experiments to ensure sufficient photoelectrons were collected. This improves the signal-to-noise ratio, ensuring that the collected spectra have enough resolution for accurate and reliable analysis.

The obtained spectra can be divided into two regions: the low binding energy region (0–5 eV) and the high binding energy region (5–17 eV). From the analysis of Figure 8, we can extract information about the density of states, as discussed below.

The density of states for the CdS and TiO_2_ P25 samples are different as expected from their different chemical composition and bonding. However, the presence of platinum produces important changes in both materials. The addition of platinum results in a greater density of states in the energy levels near the valence band, which is particularly notable in the Pt/CdS material. According to the literature, this increase is attributed to the ability of platinum 4f orbitals to donate electrons to nearby elements or materials [63]. Furthermore, the strong interaction between Pt and S observed in XPS indicates that Pt species tend to form Pt(II) and Pt(IV) rather than Pt^0^, explaining the significant increase in electron density near the upper section of the VB of the Pt/CdS composite. This is clearly reflected by the increased intensity detected for binding energies within the 5–10 eV range for Pt-containing samples. Considering that these energy levels are at the top of the valence band, the electrons in this region will be closer to the conduction band and, therefore, will be the most active in photocatalytic reactions. This interpretation of the results will allow us to understand the differences observed in the photocatalytic activity of the samples, as discussed in Section 3.2.

Additionally, further analysis of the UPS spectra allows for the construction of the absolute electronic band structure of both materials. In this context, the analysis of Figure 8b focuses on the low binding energy region (1.5–4.5 eV), as it provides detailed information about the density of states at the top of the valence band. The dashed lines in Figure 8b (see enlarged area in the figure) show the energy position of the intersection of a straight line fitted to the first region of the spectrum. The literature recommends using the second derivative of the UPS spectrum for more precise results [64]. However, for CdS, studies of both the first and second derivatives of the UPS spectrum do not provide a clear analysis of this region. In these solids, the low-energy region in both the first and second derivatives of the UPS spectrum is diffused, making it difficult to determine the exact position of the valence band maximum. In contrast, the spectrum for Titania is more evident and offers clearer results in this aspect [65].

UPS spectra were measured using a silver foil with a work function of 4.2 eV as a reference. Therefore, the valence band maximum can be calculated once the work function value is corrected. The energy values obtained from the intersections of the UPS spectra (E_VBmax_) correspond to energy values below the vacuum energy level and are thus negative. These energy values, corrected for the work function (E_VB_ + Eϕ_Ag_), are presented in Table 10. Using the bandgap energy E_g_, the energy value for the LUMO in the conduction band minimum (E_CB_) can be determined (Table 10).

Figure 9 schematically shows the possible absolute electronic band structure of the materials forming the heterostructure of the synthesized Pt/Cd(X)-Ti materials. It is important to note that the energy levels are presented in both electronvolts and as a redox potential relative to the normal hydrogen electrode (NHE), as shown in other studies. The obtained results are consistent with those reported in other studies [42,66,67]. Figure 9 shows that the electrons of the heterostructure with Pt have a higher reduction potential which improve the catalytic activity of the photocatalysts as shown in Section 3.2.

To study the charge carrier dynamics of all photocatalysts, both steady-state and time-resolved photoluminescence (PL) spectroscopy were employed. Figure 10a,b shows the PL spectra of the materials before and after Pt impregnation at steady-state, respectively. In both cases, two absorption bands are observed around 400 nm (3.1 eV) and 500–600 nm (2.5–2 eV). In the case of materials containing P25, the first emission peak can be assigned to self-trapped excitons (STE) located within the TiO_6_ octahedra, implying that the peak originates from intrinsic states rather than surface states [68]. For this reason, it can be attributed to a direct transition in the Rutile and Anatase phases, which aligns well with the band-gap energy of P25 (3.0 eV for Rutile, 3.2 eV for Anatase). Thus, this band corresponds to photo absorption near the conduction band edge and reflects a bulk property of P25 [69]. It is not possible to distinguish the absorption bands of both phases due to their similar absorption energies, resulting in an overlapping peak. On the other hand, the second emission range can be attributed to oxygen vacancies in the case of titania [70] and to the direct transition of charge carriers in the case of CdS [71]. In Cd(X)-Ti powders, it is not possible to distinguish the absorption bands of both materials, resulting in an overlapping peak. Oxygen vacancies, formed due to the presence of Ti^3+^ species near the P25 surface as studied by XPS, could be responsible for this emission, which is assigned to charge transfer transitions from electrons trapped in these vacancies. Therefore, the PL band at around 500–600 nm is more sensitive to surface changes than the band around 400 nm [71]. For this reason, the addition of CdS and Pt on the P25 surface only shifts the absorption peak appearing at around 500–600 nm.

When comparing the PL intensity of Cd(X)-Ti samples (Figure 10a) in steady-state photoluminescence, a decrease in PL intensity is observed as the amount of CdS increases. This lower intensity indicates a reduced e^−^-h^+^ recombination rate on the surface of Cd(X)-Ti materials compared to P25, as the recombination rate of photogenerated electrons and holes could be inhibited due to the formation of oxygen vacancies [68,70,72]. This phenomenon is even more pronounced in platinum containing materials, as observed in Figure 10b. These results may be explained as follows: Under UV irradiation, photoinduced electrons in Pt/Cd(X)-Ti materials migrate to the surface and then transfer to Pt due to the higher work function of Pt compared to the other components [68]. After the electron transfer, the electron concentration near the surface decreases, resulting in a decrease in PL intensity. Therefore, the number of electrons transferred corresponds to the decrease in PL intensity [69,70]. Then, there are more charge carriers available in Pt/Cd(X)-Ti than in P25 for the carbon dioxide photoreduction.

Finally, the average charge carrier lifetime (Table 11) was obtained following the signal decay for the PL signal appearing at 500–600 nm. The results obtained indicate that the charge carriers in the Cd(X)-Ti materials have a longer lifetime as the amount of CdS increases. The incorporation of Pt nanoparticles further improves this lifetime. However, these lifetimes are shorter compared to those of titania. This could be due to the fact that, despite the apparent formation of new oxygen vacancies in P25 from Pt/Cd(X)-Ti samples, as suggested by the PL intensity, Pt and CdS increase the number of carriers available for the reactions, enhancing their reduction potential, as observed in the UPS results. However, these carriers have a shorter lifetime, which is still sufficient for the CO_2_ reduction reaction to take place.

### 3.2. Photocatalytic Tests

The catalytic activity in the carbon dioxide reduction reaction was evaluated for TiO_2_ (P25) and the synthesized Pt catalysts (Pt/TiO_2_ and Pt/Cd(x)-Ti) by monitoring the amounts of generated CH_4_ and CH_3_CHO. Other subproducts were detected as traces and could not be quantified. Photocatalytic tests were conducted under the conditions described in Section 2.5. Additionally, blank tests in the absence of CO_2_ were performed to determine the methane and acetaldehyde produced solely from ethanol. The calculations are included in the Appendix A. The results, presented in Figure 11, indicate that the controlled amount of CdS and Pt species are necessary to effectively improve the photocatalytic activity of P25. The catalytic activity increases in all cases. From the results obtained, it is observed that CdS modifies the selectivity of the obtained products, while Pt nanoparticles enhance the reduction catalytic activity of the photocatalysts. It must be noted that, for comparison purposes, a physical mixture composed of CdS and TiO_2_ (with a 1% CdS loading) was prepared by grinding the two solids in an agate mortar. This physical mixture was then impregnated with Pt and tested in the reduction of CO_2_ as described in the Experimental section. The CH_4_ amount generated with this catalyst was far inferior than that of sample Pt/Cd(1)-Ti (316 µmol CH_4_/g·h vs. 460 µmol CH_4_/g·h). This observation indicates a more favorable interaction between both materials that boost the catalytic activity of the system when the catalyst preparation is done following the solvothermal method, which in turn ensures an intimate contact between the semiconductors. The lifetime of the electron-hole pairs can also be correlated with the selectivity of the materials for the formation of specific products in catalytic activity. According to the literature, the formation of C1 compounds such as CH_4_ and C2 compounds such as CH_3_CHO requires different recombination times for the e^−^-h^+^ pairs, with longer times needed for compounds containing a higher number of carbon atoms [8]. Therefore, methane generation was more prominent in the Pt/Cd(1)-Ti material, while an increase in acetaldehyde production was observed in the Pt/Cd(10)-Ti sample. Nonetheless, the catalytic activity results suggest that both CdS and Pt are necessary to generate more charge carriers with a higher reduction potential, which, although they have a shorter lifetime, are sufficient to enable electron transfer between the semiconductor and the reactants, as corroborated by the UPS and PL results. Moreover, in the former case, the uncoordinated Cd^2+^ species, as determined by XPS, may also be responsible for the enhanced catalytic activity compared to the Pt/P25 material. Additionally, both materials exhibited platinum metallic species on the catalyst surface, as confirmed by the TEM and XPS results. For Pt/Cd(10)-Ti and Pt/Cd(20)-Ti, platinum (II) and (IV) species tend to form, which may be more selective towards acetaldehyde production. Finally, as explained earlier, the Pt/Cd(20)-Ti photocatalyst is expected to exhibit the best catalytic activity for acetaldehyde production. However, it is less efficient than Pt/Cd(10)-Ti because an excess of CdS can hinder the reactants from reaching the titania. Therefore, under these conditions, the catalyst loses activity.

The obtained results in this work using Pt-impregnated CdS/TiO_2_ heterostructures as photocatalysts show a significant enhancement of the photocatalytic activity towards the photoreduction of CO_2_ to CH_4_, reaching an outstanding methane yield of 460 µmol/g⋅h, which is among the highest methane yields reported in the literature using flow reactors (see Table 12).

## 4. Conclusions

In this study, different CdS/TiO_2_ heterostructures with Pt species as co-catalysts were synthesized. The characterization results using XRD, ICP-OES, TEM, and SEM-EDX demonstrate that the synthesis of CdS and the impregnation of Pt nanoparticles were successful in all materials. XRD diffractograms show a decrease in the amount of crystalline phases corresponding to CdS after Pt impregnation, attributed to partial material removal. In this regard, TEM micrographs reveal modifications in the CdS particles. These techniques have allowed us to determine the size, morphology, distribution, and growth of Pt nanoparticles in the Pt/Cd(X)-Ti, with observed changes depending on the CdS content.

The results obtained from UV-Vis and UPS suggest that the Pt/CdS/TiO_2_ material could form a heterostructure with the valence and conduction bands shifted energetically compared to the material without Pt, favoring the CO_2_ reduction reaction. On the other hand, XPS confirmed a strong interaction between the Pt nanoparticles and CdS on the surface of the photocatalysts, as well as the presence of uncoordinated Cd^2+^ species that seem to enhance the catalytic activity but modifies the selectivity of the catalysts. Finally, PL results show that the average lifetime is directly related to the selectivity of the generated products, which in turn depends on the CdS content in the materials.

The catalytic activity exhibited by the catalysts for the CO_2_ reduction reaction has been satisfactory, thanks to the surface species generated after the impregnation of Pt nanoparticles, whose physicochemical properties strongly depend on the CdS proportion in the material. Therefore, this phase plays a fundamental role in the catalytic activity and can be modulated based on its content.

## Figures and Tables

**Figure 1 nanomaterials-14-01809-f001:**
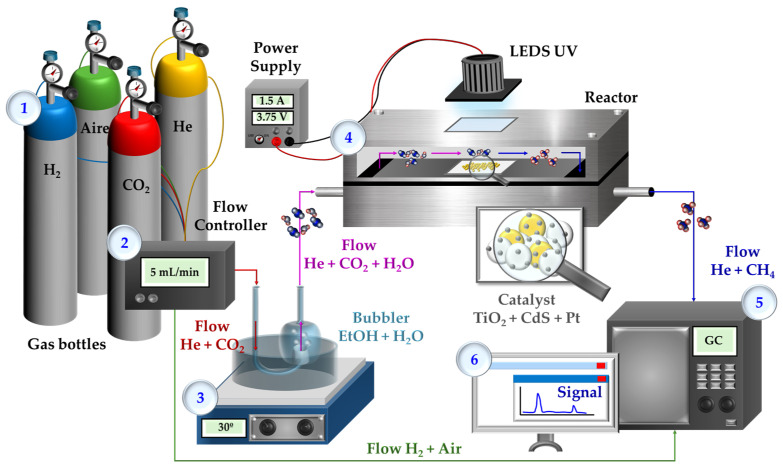
Illustration of the experimental setup and photoreactor design used for CO_2_ photoreduction catalytic tests.

**Figure 2 nanomaterials-14-01809-f002:**
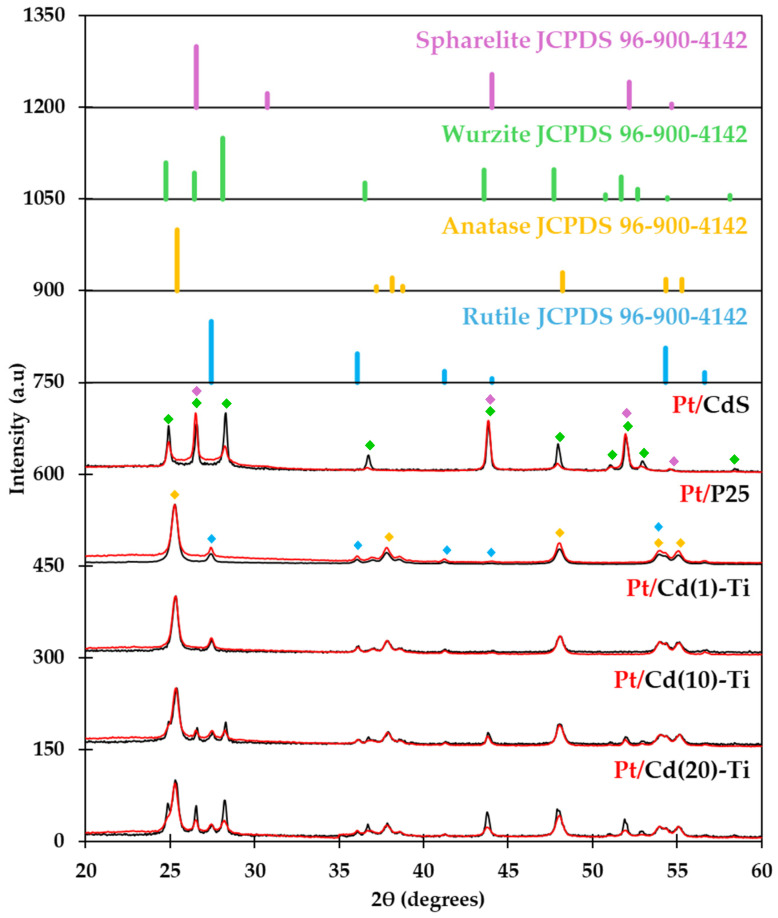
XRD patterns of the different samples and JCPDS patterns of Spharelite, Wurtzite, Anatase, and Rutile phases.

**Figure 3 nanomaterials-14-01809-f003:**
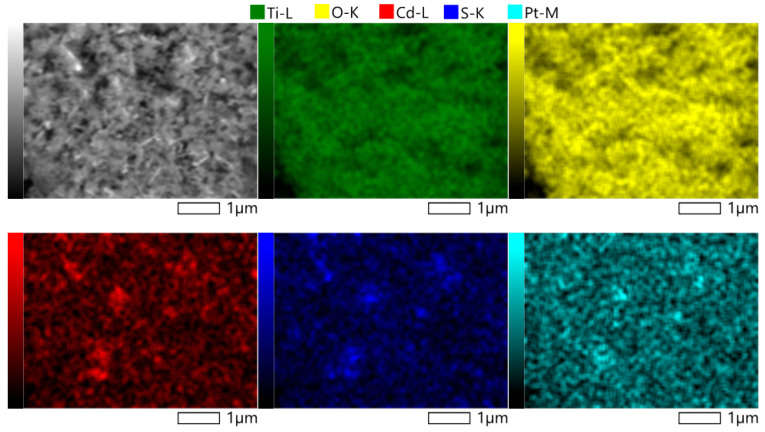
EDS elemental mapping of Cd(20)-Ti sample.

**Figure 4 nanomaterials-14-01809-f004:**
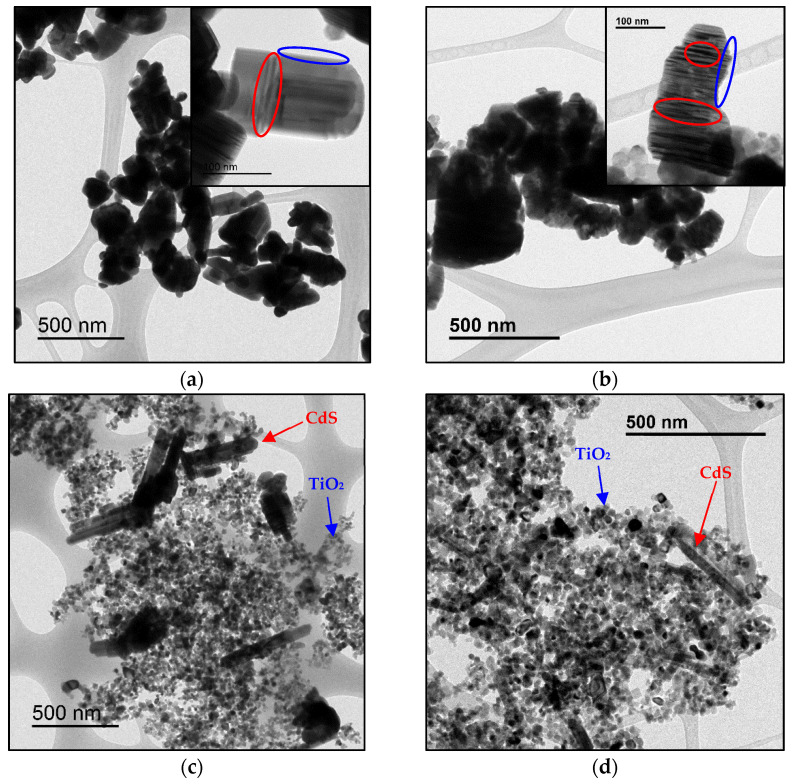
TEM images of (**a**) CdS, (**b**) Pt/CdS, (**c**) Cd(20)-Ti, and (**d**) Pt/Cd(20)-Ti samples.

**Figure 5 nanomaterials-14-01809-f005:**
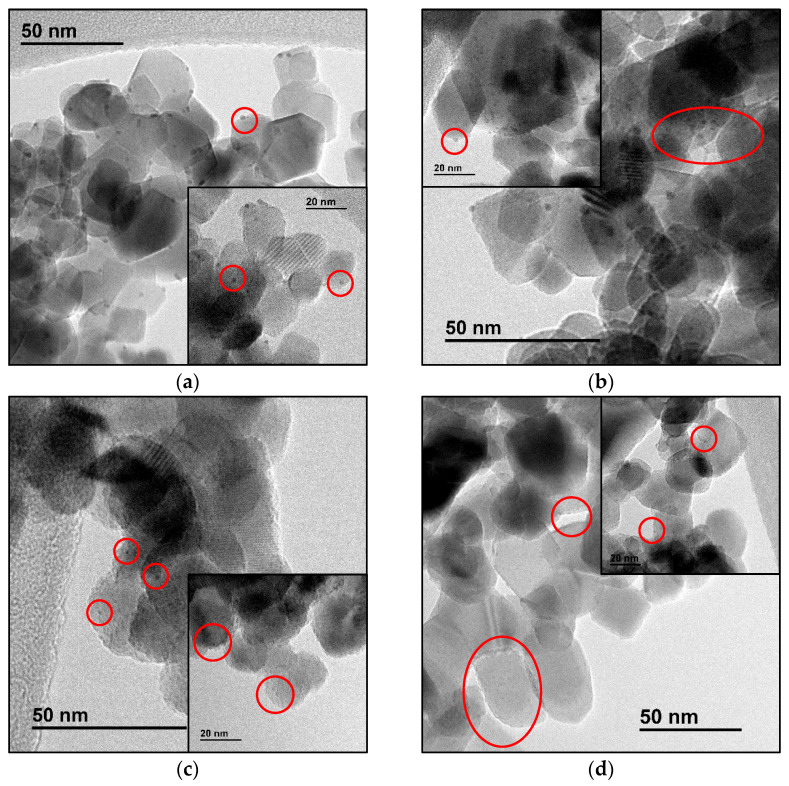
TEM images of Pt nanoparticles in (**a**) Pt/P25, (**b**) Pt/Cd(1)-Ti, (**c**) Pt/Cd(10)-Ti, and (**d**) Pt/Cd(20)-Ti samples.

**Figure 6 nanomaterials-14-01809-f006:**
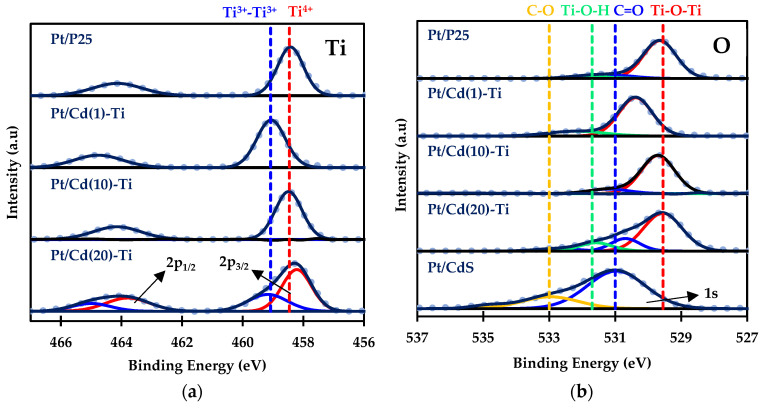
XPS spectra of (**a**) Ti, (**b**) O, (**c**) Cd, (**d**) and (**e**,**f**) Pt elements.

**Figure 7 nanomaterials-14-01809-f007:**
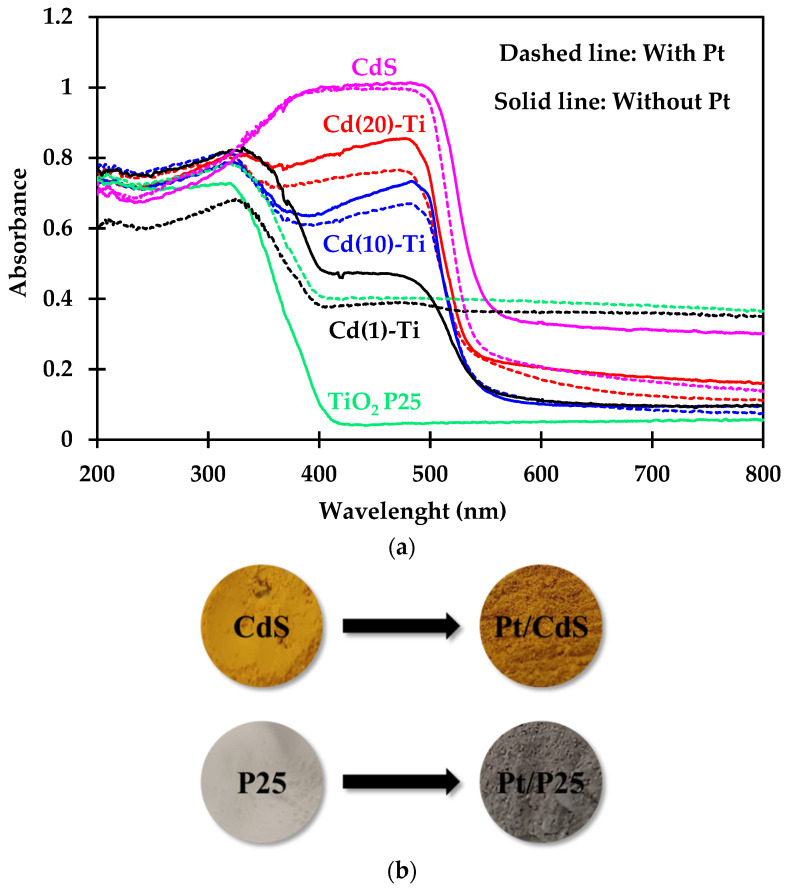
(**a**) UV-Vis spectra, all materials. (**b**) Color change in CdS and P25 materials.

**Figure 8 nanomaterials-14-01809-f008:**
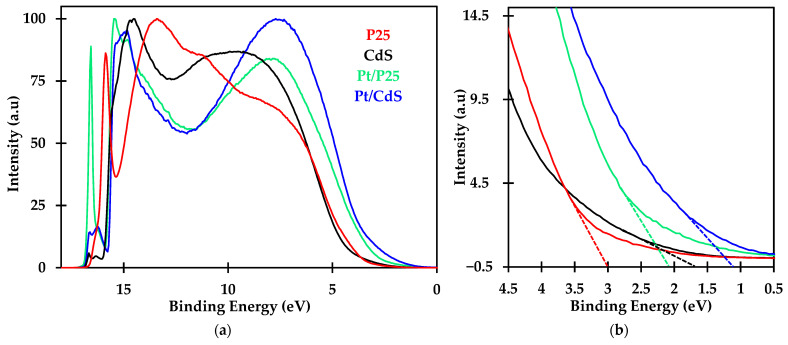
(**a**) UPS spectra of pure and platinum CdS and P25 materials. (**b**) Energy position of the intersection of a straight line fitted to the first region of the spectrum.

**Figure 9 nanomaterials-14-01809-f009:**
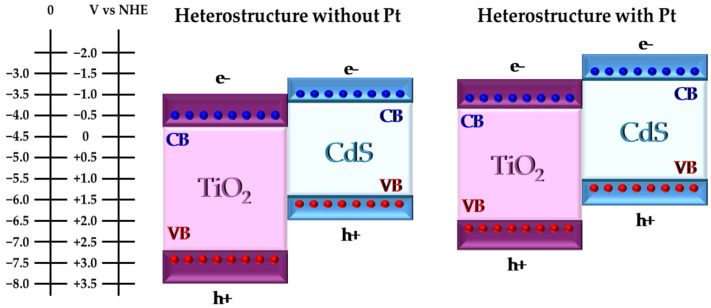
Scheme of VB and CB of Cd(X)-Ti and Pt/Cd(X)-Ti materials.

**Figure 10 nanomaterials-14-01809-f010:**
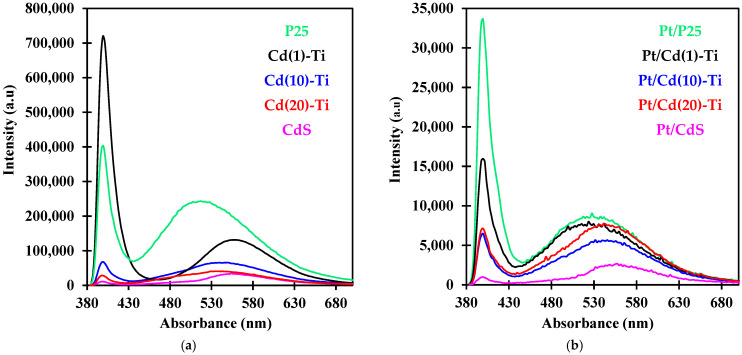
(**a**) Steady-state PL spectra of P25, CdS, and Cd/(X)-Ti materials. (**b**) Steady-state PL spectra of Pt/P25, Pt/CdS, and Pt/Cd/(X)-Ti materials.

**Figure 11 nanomaterials-14-01809-f011:**
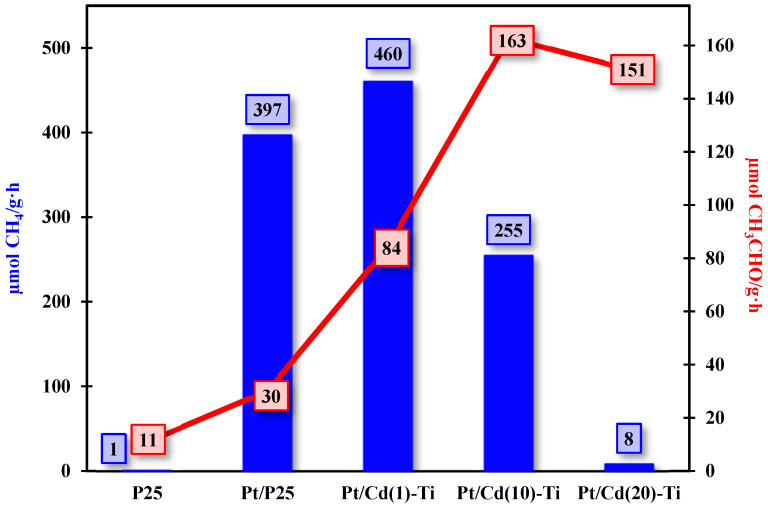
Catalytic activity in the carbon dioxide reduction reaction evaluated for the synthesized Pt catalysts.

**Table 1 nanomaterials-14-01809-t001:** Mass of reagent (in grams) required for each synthesis.

Synthesis	Photocatalysts	% CdS/TiO_2_	Cd(NO_3_)_2_·4H_2_O (g)	Na_2_S_2_O_3_ (g)	PVP10-100G (g)	TiO_2_ (g)
1	CdS	100/0	2.0927	2.1452	0.5538	0
2	Cd(20)-Ti	20/80	0.4185	0.4290	0.1108	0.8000
3	Cd(10)-Ti	10/90	0.2093	0.2145	0.0554	0.9000
4	Cd(1)-Ti	1/99	0.0209	0.0214	0.0055	0.9900

**Table 2 nanomaterials-14-01809-t002:** Nomenclature of synthesized photocatalysts.

Synthesis	% CdS	Catalyst Without Pt	Catalyst with 1% Pt
1	100	CdS	Pt/CdS
2	20	Cd(20)-Ti	Pt/Cd(20)-Ti
3	10	Cd(10)-Ti	Pt/Cd(10)-Ti
4	1	Cd(1)-Ti	Pt/Cd(1)-Ti
5	0	TiO_2_-P25	Pt/P25

**Table 3 nanomaterials-14-01809-t003:** Planes and angles (in degrees) of the different crystalline phases.

Shape	Material	Crystalline Phase	2θ (Degrees)	Plane
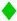	CdS	Wurtzite [50]	24.7, 26.6, 28.2, 36.8, 43.8, 48, 51, 52, 53, 55, 58.3	(100) (002) (101) (102) (110) (103) (200) (112) (201) (004) (202)
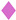	Spharelite [51]	26.6, 43.8, 52	(111) (220) (311)
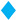	TiO_2_	Rutile [52]	27.4, 36.1, 41.2, 44, 54	(110) (101) (111) (210) (211)
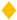	Anatase [52]	25.3, 37.8, 48, 54, 55.1	(101) (004) (200) (105) (211)

**Table 4 nanomaterials-14-01809-t004:** Percentage (%) of crystalline phase present in the materials.

Material	Anatase %	Rutile %	TiO_2_ %	Wurtzite %	Spharelite %	CdS %
CdS	0	0	0	85.1	14.9	100
Pt/CdS	0	0	0	56.6	43.4	100
TiO_2_-P25	85	15	100	0	0	0
Pt/P25	84.5	15.5	100	0	0	0
Cd(1)-Ti	82.3	17.1	99.4	0.3	0.3	0.6
Pt/Cd(1)-Ti	81.1	18.7	99.8	0.1	0.1	0.2
Cd(10)-Ti	74.8	15.2	90	9.8	0.2	10
Pt/Cd(10)-Ti	80.9	15.1	96	3.3	0.7	4
Cd(20)-Ti	67.6	14.3	81.9	16.8	1.3	18.1
Pt/Cd(20)-Ti	72	14.9	87	12.3	0.7	13.1

**Table 5 nanomaterials-14-01809-t005:** Crystallinity average particle size of CdS and P25 from XRD results.

Material	CdS Nanoparticle Size (nm)	P25 Nanoparticle Size (nm)
CdS	68	-
Pt/CdS	48	-
TiO_2_-P25	-	21
Pt/P25	-	23
Cd(1)-Ti	-	23
Pt/Cd(1)-Ti	-	23
Cd(10)-Ti	107	23
Pt/Cd(10)-Ti	59	23
Cd(20)-Ti	52	20
Pt/Cd(20)-Ti	20	23

**Table 6 nanomaterials-14-01809-t006:** Percentage of Pt present in CdS, P25, and Pt/Cd(X)-Ti samples.

Material	Pt %
Pt/CdS	1.02
Pt/Cd(20)-Ti	0.97
Pt/Cd(10)-Ti	0.60
Pt/Cd(1)-Ti	0.61
Pt/P25	0.67

**Table 7 nanomaterials-14-01809-t007:** Mass and atomic percentage of elements present in Pt/Cd(20)-Ti.

Element	Line	Mass %	Atom %
C	K	15.24 ± 0.02	29.99 ± 0.04
O	K	31.48 ± 0.06	46.52 ± 0.09
S	K	2.96 ± 0.03	2.18 ± 0.02
Ti	L	38.30 ± 0.28	18.90 ± 0.14
Cd	L	10.61 ± 0.15	2.23 ± 0.03
Pt	M	1.42 ± 0.08	0.17 ± 0.01

**Table 8 nanomaterials-14-01809-t008:** Average sizes of Pt nanoparticles.

Material	Pt Nanoparticles TEM Size
Pt/P25	2.2 ± 0.5
Pt/Cd(1)-Ti	1.4 ± 0.3
Pt/Cd(10)-Ti	1.1 ± 0.3
Pt/Cd(20)-Ti	1.1 ± 0.3

**Table 9 nanomaterials-14-01809-t009:** Experimental values of band gap (in eV) of pure and platinum CdS and P25 samples.

Material	Wavelength (nm)	Band Gap (eV)
CdS	552	2.24
Pt/CdS	546	2.27
TiO_2_-P25	414	2.99
Pt/P25	470	2.64

**Table 10 nanomaterials-14-01809-t010:** Energy levels (in eV) of the band positions.

Material	E_VB_ (eV)	E_VB_ + Eϕ_Ag_ (eV)	E_g_ (eV)	E_CB_ (eV)
CdS	−1.71	−5.91	2.24	−3.67
Pt/CdS	−1.22	−5.42	2.27	−3.15
TiO_2_-P25	−3.05	−7.25	2.99	−4.26
Pt/P25	−2.22	−6.42	2.64	−3.78

**Table 11 nanomaterials-14-01809-t011:** Fluorescence lifetime measured over the reference and Pt/CdS/TiO_2_ materials.

Material	A_1_	A_2_	A_3_	τ_1_ (ns)	τ_2_ (ns)	τ_3_ (ns)	t (ns)
CdS	3.29	0.38	3.88	2223	6764	634	1635
Pt/CdS	0.30	3.04	4.26	7057	2391	740	1650
P25	0.220	4.93	2.54	14,870	1045	3860	2370
Pt/P25	2.63	4.78	0.205	3690	1030	14,850	2319
Cd(1)-Ti	3.52	0.44	3.43	515	4803	1715	1327
Pt/Cd(1)-Ti	2.57	0.248	4.38	3026	10,625	888	1987
Cd(10)-Ti	0.362	3.44	3.83	8251	2301	660	1760
Pt/Cd(10)-Ti	0.257	2.96	4.17	9485	2698	797	1861
Cd(20)-Ti	2.54	4.44	0.283	3140	902	10,518	2059
Pt/Cd(20)-Ti	0.179	2.62	4.86	1290	3370	914	2034

**Table 12 nanomaterials-14-01809-t012:** CO_2_ reduction production outputs for gas–solid phase reactions in different photoreactors reported in the literature.

Photocatalysts	Light Source	Condition	Main Products	Ref.
CdS-TiO_2_ Nanocomposite	125 W Hg lamp 350 nm < λ < 400 nm	Type reactor: Batch reactor system	0.1875 µmol CH_4_/g·h 1.25 µmol CO/g·h	[73]
Pt/TiO_2_/SiO_2_	UV 6 W lamps λ < 365 nm 71.7 W/m^2^	Type reactor: Continuous-flow mode in a stainless-steel reactor	16.67 µmol CO/g·h 11.11 µmol CH_4_/g·h 55.56 µmol H_2_/g·h	[74]
Pt-{101]/{001}TiO_2_	300 W Xe lamp 300 nm < λ < 400 nm 20.5 mW·cm^−2^	Type reactor: Internal circulated Pyrex glass reactor	5 µmol CH_4_/g·h 10 µmol H_2_/g·h	[75]
CdSe/Pt/TiO_2_	300 W Xe arc lamp	Type reactor: Crucible inside a stainless-steel cube	48 ppm CH_4_/g·h 3.3 ppm CH_3_OH/g·h Trace amounts of CO and H_2_	[76]
3D ordered macroporous TiO_2_-supported Pt@CdS core-shell nanoparticles	300 W Xe lamp 320 < λ < 780 nm 100 mW·cm^−2^	Type reactor: Gas-closed circulation system Flow: 15 mL/min	16.2 µmol H_2_/g·h 98.7 µmol O_2_/g·h 0.7 µmol CO/g·h 36.8 µmol CH_4_/g·h	[32]
P25	UV lamp λ = 365 nm 75 mW·cm^−2^	Type reactor: Ring continuous-flow reactor Flow: 0.063 mL/min.	1.3 μmol CO/g·h 1.2 μmol CH_4_/g·h	[77]

## Data Availability

Data are contained within the article and Appendix A.

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
