# Peer review of "Exploring Pt-Impregnated CdS/TiO2 Heterostructures for CO2 Photoreduction"

_nanomaterials, 2024, doi:10.3390/nano14221809_

Round 1

Reviewer 1 Report

Comments and Suggestions for Authors

The authors have investigated methane production through the photocatalytic reduction of carbon dioxide using CdS/TiO₂ heterostructures doped with platinum (Pt). The study describes the preparation of the photocatalysts, utilizing commercial titania (P25) and CdS synthesized through a solvothermal method, followed by Pt impregnation on the surface to enhance the physicochemical properties of the resulting photocatalysts. The materials were characterized by various techniques, including XRD, ICP-OES, SEM, TEM, and XPS, among others. The characterization section, focusing on the structural properties and understanding the material’s framework, is comprehensive. However, the catalytic performance was not thoroughly explored to fully understand the behavior of the catalyst and which specific properties influence its activity. Overall, the manuscript holds scientific merit, presenting interesting and promising results. Therefore, approval is recommended with some revisions.

Clarification Regarding the Presence of Pt in the Material: It is recommended to clarify whether platinum (Pt) is present in the material as a dopant or if it is impregnated on the surface. It is observed that Pt is impregnated on the surface of the material rather than functioning as a dopant. Therefore, it is suggested that the title of the work be modified to reflect this distinction.

Suggestion Regarding Figures 1 and 2: Figures 1 and 2 do not appear to be essential to the main manuscript, as they do not present significant innovation concerning the described synthesis processes. It is recommended that they be transferred to the supplementary material.

Correction in Figure 3 (Experimental Setup): Is the illustration of the experimental setup in Figure 3 correct? Do all gases pass through the reactor, or do only He and CO₂ pass through while the other gases go directly to the chromatograph? A review of this section is recommended.

Interpretation of TEM Images: The TEM images indicate that there is no heterostructure formed between TiO₂ and CdS, but rather a mixture of the materials, likely due to the size of the formed CdS particles. The heterostructure appears to be observed only in the presence of Pt (Figure 7). Therefore, it is suggested to conduct a test involving a mechanical mixture of the two materials to verify the actual efficacy of the synthesis process and confirm whether there is indeed a junction between the materials that benefits the photocatalytic reaction.

Modification of Figure 9: It is suggested to remove Figure 9b or, alternatively, to include a real image of the materials, possibly as an inset in Figure 9a.

XPS Data Treatment: The authors are advised to review the treatment of XPS data for the Pt/Cd(1)-Ti material to ensure that there were no calibration errors in the high-resolution spectrum, as the peaks are quite similar to those of the Pt/Cd(10)-Ti and Pt/P25 samples. The shifts observed in the high-resolution spectra of Ti2p and O1s may be due to errors in data processing.

Band Gap Calculation: It is recommended to review the methodology used for calculating the band gap value to verify its appropriateness and accuracy.

Presentation of Selectivity Data: It is advised to include a graph of product selectivity according to the material used or to present this data in detail within the text, also in accordance with the materials used.

Absence of Catalytic Tests with TiO₂ (P25): It would be important to clarify why catalytic tests were not conducted exclusively with TiO₂ (P25), as this information may be relevant for comparing the results.

Reviewer 2 Report

Comments and Suggestions for Authors

In this work, the authors present the results of synthesis of TiO2/CdS doped Pt nanoheterostructures. The structure of the obtained materials, as well as the interaction of their components, have been investigated using a wide range of modern methods. The selectivity of the catalytic action of the obtained catalysts in the processes of CO2 reduction to methane, as well as CH3CHO depended on the ratio of components was found. The paper is well structured and contains a sufficient amount of data characterizing the structure of the obtained nanomaterials. Nevertheless, there are a number of remarks to the paper, the elimination of which will improve the quality of the manuscript.

1. To date, TiO2/CdS/Pt structures are quite common objects. There is a great number of works devoted to the synthesis of these nanomaterials, including those doped with Pt and used as photocatalysts for various chemical processes. However, in the introduction, the authors almost do not discuss these works, and the reader has the feeling that the authors have obtained these nanomaterials for the first time. The authors should analyze similar works and highlight the advantages of their proposed approach.

2. In the introduction, the authors indicate that the content of anatase and rutile phases in P25 is 70%:30%. Whereas in the experimental part for the same material the rutile:anatase ratio is 85%:15%.

3. In Table 1, the designation of Cd(X)-Ti materials is missing.

4. In the section describing the synthesis, there is no description of the role of PVP in the synthesis of the materials.

5. Arrows describing the directions of the process should be added to Figures1 and 2

6. Scherrer's formula is very often used. There is no need to keep citing it in the text and it is better to limit it to a reference to a previously published work.

7. In TEM images presented in the figure clearly visualized nonspherical particles whose size varies in the range of 30-80 nm. The authors need to explain the nature of these particles. 

8. The sentence ........In the case of materials containing P25, the first emission peak 500 can be assigned to self-trapped excitons (STE) located in the TiO6 octahedra, implying that  the peak originates from intrinsic states rather than surface states [64] ..........requires correction.

Reviewer 3 Report

Comments and Suggestions for Authors

In this manuscript, a heterojunction Pt-doped CdS/TiO2 photocatalyst was synthesised using solvothermal process followed by impregnation method, and its photocatalytic performance was further studied. Some points need to be clarified:

1.      The language used in some parts of introduction does not sound natural as it is quoted from page 1, line number 40-41; ‘Carbon dioxide has the potential to support the development of products and services with a reduced carbon footprint thanks to its unique physicochemical properties.’ I recommend to revise the statement (s).

2.      For Fig.1 and 2, I recommend to include arrows to show the flow of synthetic steps of the photocatalysts.

3.      To confirm the peaks of phases formed whether they are matching accordingly, I recommend the authors to include the JCPDS peaks in the XRD graph.

4.      Fig. 9 (b) should be replaced with the captured images of samples collected, otherwise only Fig. 9 (a) is enough to convey the information intended.

5.      According to statements made in page 16, line number 455-459; the author stipulated that the interaction between Pt and S in Pt/CdS has led to increased electron density near VB and the corresponding electrons have been raised near CB, the authors should clarify this phenomenon, why the band gap of Pt/CdS (2.27 eV) is slightly higher than CdS (2.24 eV).

6.      The authors are recommended to compare intensively their work in relation to TiO2-based heterostructures employed in CO2 photoreduction.
